# Porous Polymer Materials for CO_2_ Capture and Electrocatalytic Reduction

**DOI:** 10.3390/ma16041630

**Published:** 2023-02-15

**Authors:** Hui Wang, Genyuan Wang, Liang Hu, Bingcheng Ge, Xiaoliang Yu, Jiaojiao Deng

**Affiliations:** 1Key Laboratory for Green Chemical Technology of Ministry of Education, Haihe Laboratory of Sustainable Chemical Transformations, School of Chemical Engineering and Technology, Tianjin University, Tianjin 300072, China; 2State Key Laboratory of Physical Chemistry of Solid Surfaces, Collaborative Innovation Center of Chemistry for Energy Materials, College of Chemistry and Chemical Engineering, Xiamen University, Xiamen 361005, China; 3Department of Mechanical Engineering, Research Institute for Smart Energy, The Hong Kong Polytechnic University, Hong Kong, China; 4Graphene Composite Research Center, College of Chemistry and Environmental Engineering, Shenzhen University, Shenzhen 518060, China

**Keywords:** polymer, porous structure, CO_2_ capture, CO_2_ reduction

## Abstract

Efficient capture of CO_2_ and its conversion into other high value-added compounds by electrochemical methods is an effective way to reduce excess CO_2_ in the atmosphere. Porous polymeric materials hold great promise for selective adsorption and electrocatalytic reduction of CO_2_ due to their high specific surface area, tunable porosity, structural diversity, and chemical stability. Here, we review recent research advances in this field, including design of porous organic polymers (POPs), porous coordination polymers (PCPs), covalent organic frameworks (COFs), and functional nitrogen-containing polymers for capture and electrocatalytic reduction of CO_2_. In addition, key issues and prospects for the optimal design of porous polymers for future development are elucidated. This review is expected to shed new light on the development of advanced porous polymer electrocatalysts for efficient CO_2_ reduction.

## 1. Introduction

The usage of fossil energy has enabled human society to prosper, but it has also led to serious environmental problems [1]. Since the burning of coal, petroleum, and natural gas, the CO_2_ fixed by plants from nature through millions of years of photosynthesis has been released by humans in just a few hundred years [2]. Moreover, the greenhouse effect caused by excessive CO_2_ emissions contributes to the climate crisis, sea level rise, and ocean acidity increase [3,4,5]. Up to now, “zero carbon” or “carbon neutral” climate goals have been set up in more than 130 countries or regions [6]. In order to achieve this goal, research work focuses on finding clean energy alternatives to fossil fuels to reduce CO_2_ emissions, and developing technologies such as CO_2_ capture, storage, utilization, and chemical conversion to reduce the concentration of CO_2_ in the air [7,8,9].

The International Energy Agency predicted that CO_2_ capture and storage (CCS) can reduce CO_2_ emissions by about 14% annually, which is an important green “bridging” strategy in greenhouse gas reduction [10]. CO_2_ capture primarily includes pre-combustion, oxyfuel combustion, and post-combustion. CO_2_ sequestration is the last link of the CCS chain, which focuses on marine and geological sequestration and mineral carbonation [11]. In addition, the research focus of CO_2_ capture lies on post-combustion technology, which can be applied to the capture of low-pressure CO_2_ in flue gases [12]. In industry, CO_2_ in mixed exhaust gases can be effectively captured by amine solutions such as monoethanolamine (MEA) and diethanolamine (DEA). However, the high degree of toxicity, corrosion risk, as well as intense energy consumption hinder their wide applications [13]. It is necessary to develop environmental-friendly CO_2_ capture materials when considering people’s health and gradually serious environmental problems [14,15]. Morphology, functionality, structure stability, and pore structure are key parameters that determine the efficiency of an adsorbent. In particular, the size of the pore regulates the physical/chemical interaction between the pore and the gas. Porous materials with a pore size closer to the kinetic diameter of CO_2_ have a higher CO_2_ affinity. Typically, microporous materials show excellent adsorption performance at lower CO_2_ pressures [16], while mesoporous materials perform better for CO_2_ adsorption at higher pressures [17]. The equilibrium adsorption capacity of adsorbent varies with pressure or temperature. In industry, CO_2_ regeneration is mainly achieved by reducing pressure or increasing temperature [18,19].

From another perspective, CO_2_ is a low-cost, atoxic, and renewable carbon resource. So, the achievement of CO_2_ capture and conversion not only mitigates the greenhouse effect but also reduces human industry’s dependence on fossil fuels. However, CO_2_ has a low standard Gibbs free energy ΔG_0_ = −394.38 kJ mol^−1^. Moreover, CO_2_ shows kinetic inertia and thermodynamic stability, which make it difficult to be activated [20]. Therefore, the transformation of CO_2_ generally requires reactions with high-energy molecules (e.g., H_2_, epoxides, unsaturated complexes, organometallic complexes) or relies on external energy (e.g., thermal energy, solar energy, or electricity). In the above strategy, electrochemical technology enables an efficient conversion between electrical and chemical energy. CO_2_ electrocatalytic reduction is valued due to its mild reaction conditions and renewable clean energy source [21]. Developing novel electrocatalysts, atomic engineering at the catalysts’ surface interface and increasing the specific surface area all contribute to the boosting the catalytic activity of the electrocatalysts [22,23,24]. In addition, a large number of metallic (e.g., metals, alloys, metal oxides, metal carbides, metal sulfides, etc.), non-metallic (e.g., carbon nanotubes, graphene, polymers), and combinations of them served as efficient electrocatalysts for different electrocatalytic reactions [25,26,27]. In previous studies, porous materials have been shown to have more accessible catalytic sites, larger specific surface areas, rich structural designs, and suitable constraints on active species for superior catalytic performance compared to non-porous materials. Therefore, porous materials have the potential to be applied to CO_2_ capture and electrocatalysis (Figure 1). Moreover, the functional modules of CO_2_ capture and electrocatalytic reduction can be integrated in a single system, where the porous material adsorbs CO_2_ and subsequently completes the catalytic reaction process at the catalytic active site to directly produce a chemical with high added value (e.g., methanol, acetic acid). Recently, interesting research advances have been made in CO_2_ capture and electrocatalytic reduction, involving porous materials such as zeolites [28], porous carbon materials [29], and porous polymers [30].

Among them, porous polymers are particularly attractive due to their high specific surface area, regulatable porosity, and precisely designed active sites. Some previous reviews have covered the studies of porous polymers in the field of CO_2_ capture and conversion [31,32], but a thorough overview of the structure–activity relationship, especially the effect of porosity on the capture capacity and catalytic activity of CO_2_ is still lacking. In this review paper, the porosity regulation in porous polymers and the influence of pore sizes on CO_2_ capture and electrocatalytic reduction are comprehensively summarized. Special attention will be paid to the interaction between pore structure with CO_2_ and reaction intermediates. Porous polymers as CO_2_ adsorbents and catalysts for CO_2_ electrocatalytic reduction will be discussed in this paper. Moreover, the existing limitations, the challenges in porous polymer design, and the future research direction have also been prospected.

## 2. Porous Polymers for CO_2_ Capture and Electrocatalytic Reduction

Porous polymer materials used for CO_2_ capture and electrocatalytic reduction are mainly divided into four categories, namely, porous organic polymers (POPs), porous coordination polymers (PCPs), covalent organic frameworks (COFs), and nitrogen-containing polymers (N-polymers). Such porous polymeric materials can be prepared through a variety of synthetic methods, including solvothermal, ionothermal, microwave, mechanochemical, and interfacial synthesis methods. Among them, the solvothermal method is the most common preparation method [33]. The advantage of the solvent thermal method is that it can improve the synthesis efficiency by changing the solvent, but the biggest problem is that the solvent used is mostly organic solvent, which will cause environmental pollution when used in large quantities, and the reaction conditions are harsh, so it is not suitable for large-scale production. The ionothermal method is mainly used for the synthesis of covalent triazine frameworks (CTFs), and the reaction conditions required are more severe than those of the solvent thermal method, so the ionothermal method is not suitable for industrial applications [34]. The microwave method can effectively reduce the reaction time and maintain the porosity by microwave, which is a promising synthesis method [35]. The mechanochemical method does not require solvents and therefore avoids environmental pollution and is a good green synthesis method because of the simplicity of the process [36].

The CO_2_ Reduction Reaction (CO_2_RR) generally takes three steps: CO_2_ adsorption, electrocatalytic reduction, and desorption. To be specific, gaseous CO_2_ molecules are first dissolved in the solvent and form a CO_2_-saturated electrolyte. Then, the dissolved CO_2_ in the solvent reacts on the electrode surface. Finally, the reaction products are desorbed from the electrode surface to complete the whole reaction process. Because of the large number of products and complex intermediates in multi-step reactions of CO_2_RR, definite reaction mechanisms remain unclear. Currently reported CO_2_RRs occurring in aqueous solutions are summarized in Table 1. From the reaction equations in the table, it can be seen that CO_2_ can undergo different degrees of electrochemical reduction reactions by transferring different numbers of electrons, and the products include HCOOH, CH_3_OH, CO, CH_4_, CH_3_COOH, C_2_H_4_, CH_3_CH_2_OH, C_2_H_6_, etc. Additionally, the reduction of CO_2_ is accompanied by HER on the cathode.

### 2.1. Porous Organic Polymers (POPs)

Porous organic polymers (POPs) are 2D or 3D network structures expanded by organic groups or fragments generated through chemical polymerization [37,38]. With a large specific surface area, variable porosity, excellent thermal stability, as well as controllable functionality, POPs are widely applied in gas adsorption, heterogeneous catalysis, energy storage, and other fields [36,39,40,41,42]. A large specific surface area and abundant porosity lead to high CO_2_ absorption capacity, bringing out the importance of pore regulation in the design of CO_2_ capture materials (Table 2). 

In recent years, many studies have focused on POPs serving as hosts or heteroatom catalysts to capture and convert CO_2_ [31,43,44,45]. Han et al. [46] obtained 1,3-dialkyne linked porous polymer LKK-CMP-1 through 1,3,6,8-Tetraethynylpyrene homopolymerization for CO_2_ capture. The N_2_ adsorption/desorption isotherm for LKK-CMP-1 is a typical I-type, indicating a microporous structure (Figure 1a). The pore size of LKK-CMP-1 is mainly distributed around 0.59 nm (Figure 1b), close to the dynamic diameter of CO_2_ (0.33 nm). The CO_2_ isosteric heat (Q_st_) for LKK-CMP-1 was 35.0 kJ mol^−1^ (Figure 1c), which proves CO_2_ molecules have a strong interaction with LKK-CMP-1 pore walls. The CO_2_ adsorption capacity can reach 9.78 wt% at 273 K (1 bar). Song et al. [47] tuned the size of micropores through side-chain engineering. Crosslinkable dianhydride monomers PEPHADA and PEQDA were utilized to construct microporous networks (Figure 1d). The crosslinkable alkyne group on the PDQDA side branch is closer to the molecular skeleton, which adjusts the pore size of porous polymers to 0.57 nm (Figure 1e). Therefore, the pore size of crosslinked polyimide can be precisely regulated by adjusting the position of the crosslinkable group in polyimide. The 6FA-PE-CL with well-developed micropores shows a higher surface area and stronger CO_2_ uptake ability. At 273 K and 1 bar, the CO_2_ uptake ability of 6FA-PE-CL is 8.9 wt% (Figure 1f). 6FA-PE-CL exhibits a significant Q_st_ value downward trend compared to 6FA-PEPH-CL with the increase of CO_2_ adsorption, which indicates that there is a stronger interaction between ultramicropores (0.56 nm) and CO_2_ molecules. 

**Table 2 materials-16-01630-t002:** The textural features and CO_2_ adsorption characteristics of reported POPs.

POPs	S_BET_ (m^2^ g^−1^)	V_total_ (cm^3^ g^−1^)	CO_2_ Uptake (mmol g^−1^) 273 K	CO_2_ Q_st_ (kJ mol^−1^ )	CO_2_/N_2_ Selectivity 273 K	Ref
isox-CTF-500	1683	0.70	4.92	29	29	[43]
acac-CTF-500	1556	1.20	3.30	28	46	[43]
cCTFs-400	744	0.36	2.86	49	-	[45]
cCTFs-450	861	0.59	2.25	46	-	[45]
cCTFs-500	1247	1.04	3.02	43	-	[45]
LKK-CMP-1	467	-	2.24	35.0	44.2	[46]
6FA-PEPH-CL	653	0.37	1.65	-	38.0	[47]
6FA-PE-CL	698	0.46	2.02	-	58.0	[47]
TTF-1	1234	0.59	2.86	-	-	[48]

POP is an ideal porous structure, which can be applied for the electrocatalytic reduction of CO_2_ through functional design. Dai et al. [48] reported a porous triazine linking framework TTF-1 with a pyridine linker as the reactive site (Figure 2a). The existence of a well-developed porous structure enables a large contact area between electrodes and electrolytes, and facilitates the charge and mass transfer (Figure 2b). Amorphous porous polymer POP often shows a broad pore size distribution, which is attributed to the inter-particle void. In particular, a large number of micropores promote the adsorption of CO_2_. The permanent CO_2_ uptake of TTF-1 is up to 64.4 cm^3^ g^−1^ (298 K, 1 bar). TTF-1 also shows excellent electrocatalytic activity for CO_2_ reduction, with a Faraday efficiency of about 82%.

The size of the pore can affect the charge density of the polymer and the confinement of the pore to the active site. Tang et al. [49] synthesized a series of porous polymers, POP-Py (n) (n means the quantity of benzene rings), through the reaction of benzyl bromide and pyridine. POP-Py (n) with pore sizes from micropore to mesopore were obtained by changing the amount of benzene rings (Figure 3a). The highly efficient electrocatalyst was obtained by encapsulating cobalt meso-tetra(4-carboxyphenyl)porphyrin (CoTCPP) (Figure 3b) into POP-Py (n) by the ion exchange method, with single-molecule dispersion. POP-Py (0)/CoTCPP showed the highest FE_co_ of 83% at −0.6 V vs. RHE, which is twice as high as the activity of the pure CoTCPP molecular electrocatalytic conversion of CO_2_ to CO (Figure 3c). The high activity is caused by the monomolecular dispersion of CoTCPP in the pores of POP-Py (0), which promotes the electrochemical accessibility of active sites. Moreover, the positively charged substituted pyridine ring in POP-Py can stabilize the key reaction intermediate *COOH to accelerate the conversion of CO_2_. When a neutral benzyl ring is inserted between two pyridine rings, the positive charge on the pyridine ring will be delocalized through the conjugation effect. Hence, the interaction between POP-Py and key reaction intermediate *COOH becomes weaker. In addition, at a high overpotential, the current attenuation of POPy-Py(2) indicated the aggregation of CoTCPP (Figure 3d).

However, many POPs have poor electrical conductivity, which hinders their application in electrochemistry. Alkordi et al. [50] homogenously deposited porous organic polymer containing pyrimidine atop a graphene sheet to provide composite material PyPOP@G (Figure 4a). It shows significant CO_2_ electrochemical reduction activity (5 mA cm^−2^ at −1.6 V), which is much higher than the separated component. PyPOP has an intentional affinity for CO_2_ due to its rich microporous structure (Figure 4b), while graphene enhances the electrical conductivity of the composite. The synergistic effect of the two components promotes the electrocatalytic reduction of CO_2_. Later, Alkordi et al. [51] prepared a composite material named POP@MWCNT by synthesizing POP on multi-walled carbon nanotubes in a one-pot method. It was verified by the N_2_ physical adsorption/desorption test that the high porosity of POP@MWCNT is mainly contributed to by POP, not MWCNTs (Figure 4c). The Q_st_ of CO_2_ into the POP and the POP@G are almost the same, which confirms that the adsorption of CO_2_ by the composites mainly occurs in the pores of POP (Figure 4d). Alkordi et al. proposed post-synthetic modification of the electrode material by initial impregnation in CuBr acetonitrile solution. Impregnation of Cu ions enhances electrode catalytic activity. The current density for the POP-Cu@MWCNTs was enhanced by 75% compared to POP@MWCNTs at −1.5 V (Figure 4e).

### 2.2. Porous Coordination Polymers (PCPs)

Porous coordination polymers (PCPs), also known as metal-organic frameworks (MOFs), are crystalline materials. Such materials are composed of metal ions or metal particles with organic complexes through coordination linkages. PCPs have now become a significant category of nanoporous polymers with multiple uses in CO_2_ storage, CO_2_ separation, and CO_2_ electrocatalysis due to their organized porous structures with large surface areas and adjustable pore surface features [52,53,54,55,56]. The catalytic performance of PCPs catalysts in CO_2_ electroreduction is summarized in Table 3. Sun et al. [57] reported a two-dimensional bismuth metal-organic framework (Bi-MOF), a catalyst with a microporous structure, for the electrocatalytic reduction of CO_2_ to HCOOH. They prepared Bi-MOFs with tunable grain size and coordination environment by a simple solvothermal method, varying the reaction temperature and the type of organic linker. Figure 5d shows the crystal structure of a Bi-MOF (CAU-17) composed of a triple base linker (1,3,5-benzene tricarboxylic acid, H_3_BTC). Figure 5a shows a representative type-I N_2_ adsorption/desorption isotherm of Bi-MOF, and after measuring it was found that the average pore size was about 0.8 nm (inset of Figure 5a), indicating a remarkable microporous structure. In particular, the obtained Bi-MOF has a CO_2_ adsorption capacity of 33.0 mg_CO2_g_cat_^−1^, which is twice the CO_2_ adsorption capacity of Bi sheets under the same testing conditions (Figure 5b). The superior CO_2_ adsorption capacity of Bi-MOF is attributed to the large number of differently shaped channels in its structure, which are larger in size than the molecular diameter of CO_2_. The excellent CO_2_ uptake capacity promotes the adsorption of large amounts of CO_2_ on the surface of Bi-MOF catalysts, thus potentially contributing to higher catalytic conversion. This Bi-MOF shows large Faraday efficiency of HCOOH over a wide potential window, reaching 92.2% at −0.9 V (vs. RHE), and excellent stability over 30 h (Figure 5c). 

The same microporous structure is a nickel phthalocyanine-based conductive MOF (NiPc-Ni(NH)_4_) constructed from 2,3,9,10,16,17,23,24-octaaminonickel(II) (NiPc-(NH_2_)_8_) as reported by Cao et al. [58]. In this research, Ni-phthalocyanines (NiPc), which was built block-based, with an activated CO_2_ molecular reduction site, were incorporated into a porous essentially conductive 2D MOF NiPc-Ni(NH)_4_ for the efficient CO_2_RR towards the CO_2_ conversion in an aqueous medium. NiPc-Ni(NH)_4_ showed a 1.78 nm one-dimensional rectangular channel while running along the c-direction with a distance of 0.34 nm between the stacked two-dimensional conjugated layers in the structural model (Figure 6a). Therefore, the N atoms of the amino groups in NiPc-(NH_2_)_8_ ligands were managed to coordinate with the Ni(II) ion, forming a fully π-conjugated two-dimensional backbone in NiPc-Ni(NH)_4_. The accessible pore size of ~1.7 nm was obtained by N_2_ adsorption/desorption isotherms (Figure 6b), and this large microporous structure facilitates the diffusion and mass transfer of reactants to improve the activity of electrocatalytic reactions. Similarly, the porous NiPc-Ni(NH)_4_ has a great CO_2_ adsorption capacity (Figure 6c), which indicates that NiPc-Ni(NH)_4_ with a nitrogen-rich structure has a strong affinity for CO_2_ to enhance its electrocatalytic activity in CO_2_RR. It can be seen that NiPc-Ni(NH)_4_ nanosheets reach a high CO Faraday efficiency of 96.4% at −0.7V (vs. RHE). 

Among PCP materials, not only are microporous structures often seen, but also mesoporous structures are common for CO_2_RR catalysts. Gu et al. [59] prepared a series of imidazole (Im)-based zeolite imidazolium salt backbone Ni(Im)_2_ nanosheets of different thicknesses using the liquid phase top-down exfoliation method (Figure 7a). Zeolite imidazole frameworks (ZIFs) are a relatively stable class of MOF materials consisting of imidazole groups and metal ions. They stripped layered massive Ni(Im)_2_ ZIF to 2D Ni(Im)_2_ nanosheets of different thicknesses (5 nm, 15 nm, 65 nm, and 140 nm) and used them as CO_2_RR electrochemical catalysts. As can be seen in Figure 7b, their N_2_ adsorption/resolution isotherms are all type VI, which corresponds to a mesoporous structure. It provides a large specific surface area for the catalysts while more reactive sites are exposed, thus ensuring the high catalytic activity of this material. This catalyst was found to achieve 78.8% CO Faraday efficiency at −0.9 V (vs. RHE) with a stability of more than 14 h (Figure 7c).

Because a single porous structure is not sufficient for high performance in CO_2_ capture and electrocatalytic reduction, some researchers have attempted to create hierarchical porous materials through pore engineering strategies [60,61,62,63]. In recent years, the field of MOF has made great progress in the development of hierarchically porous materials [64,65,66,67]. In such hierarchical porous polymers, the inherent micropores of the framework contribute to a high surface area, while the mesopores and/or macropores provide channels for mass transfer and accommodate molecules larger than the micropores [68]. Martin et al. [69] reported the deposition of porous metal-organic skeletons on copper electrodes by electrosynthesis of ionic liquid templates [Cu_2_(L)] [H4L = 4,4′,4″,4‴-(1,4-phenylenebis(pyridine-4,2,6-triyl))tetrabenzoic acid]. The structure of Cu_2_(L) comprises a 3D network built around binuclear [Cu_2_(OOCR)_4_] paddlewheels with four bridging carboxylate ligands (Figure 8a,b). The porosity of this MOF was investigated by the N_2_ adsorption/desorption test at 77 K, and Cu2(L)-e exhibited a distribution between type I and type IV, indicating the presence of both mesopores and micropores. The distribution of micropores and mesopores in both materials was analyzed by Horvath Kawazoe and Barrett–Joyner–Halenda (BJH) methods, respectively (Figure 8c,d). By contrast, the reduction of micropores in Cu_2_(L)-e resulted in lower N_2_ uptake at low pressure, while the total pore volume of Cu_2_(R)-e produced by the template effect of the ionic liquid during electrolytic synthesis was 1.89 cm^3^ g^−1^, significantly larger than that of Cu_2_(L)-t (0.32 cm^3^ g^−1^), reflecting the presence of Cu_2_(L)-e-mediated pores. The presence of mesopores also facilitates the mass transfer process during the reaction. As a result, this electrode showed excellent activity for the electroreduction of CO_2_ to formic acid with a low starting potential of −1.45 V (vs. Ag/AgCl) and HCOOH Faraday efficiency reaching 90.5% at −1.80 V (vs. Ag/AgCl). Despite a similar micro- and mesoporous structure, the role of pore structure in the materials reported by Lan et al. [70] is quite different. The specific surface area of the material decreases instead after the introduction of PPy in the material, which is due to the encapsulation of PPy in the pores of MOF-545-Co. The formation of PPy occurs in the MOF channel, and after the formation of PPy, there will be a π-π interaction between the PPy molecules and TCPP molecules in the MOF structure similar to the host–object interaction, and this interaction will facilitate the charge transfer process during the reaction. Meanwhile, the results of CO_2_ adsorption measurements showed that the addition of PPy was quite beneficial for CO_2_ adsorption. As a result, the CO_2_ reduction efficiency is significantly enhanced. 

**Table 3 materials-16-01630-t003:** Summary of catalytic performance of PCP catalysts in CO_2_ electroreduction.

Catalysts	E/V vs. RHE	Major Products	FE/%	Structure	Ref.
Bi-MOF	−0.9	HCOOH	92.2	microporous	[57]
NiPc-Ni(NH)_4_	−0.7	CO	96.4	microporous	[58]
2D Ni(Im)_2_-5 nm	−0.9	CO	78.8	mesoporous	[59]
Cu2(L)-e/Cu	−1.2	HCOOH	90.5	micro- and mesoporous	[69]
PPy@MOF-545-Co	−0.8	CO	98	micro- and mesoporous	[70]
MOF-NS-Cu	−0.6	HCOOH	83.1	mesoporous	[71]
MOF-NS-Co	−0.6	CO	98.7	mesoporous	[71]
PCN-222(Cu)	−0.7	HCOOH	44.3	mesoporous	[72]
PCN-224(Cu)	−0.7	HCOOH	34.1	microporous	[72]
Al_2_(OH)_2_TCPP-Co	−0.7	CO	76	mesoporous	[73]
PcCu-O8-Zn	−0.7	CO	88	microporous mesoporous	[74]
PCN-224-NH_3_	−0.9	CO	60	microporous	[75]
ZIF-A-LD	−1.1	CO	90.57	microporous	[76]
D-P-CoPc	−0.6	CO	97	macroporous	[77]
Cu_30%_ZIF-8	−1.0	CH_4_	35.21	microporous mesoporous	[78]
p-CuL-4	−0.37	CH_3_COOH	64	microporous	[79]

### 2.3. Covalent Organic Frameworks (COFs)

Covalent organic frameworks (COFs) are a class of reticulated porous polymeric materials in which the constructed organic blocks are connected by covalent bonds. COFs are characterized by their tunable porosity, which gives them a great accessible surface area and an abundance of active sites. It is due to this property that they have also been extensively studied in catalytic and energy storage systems [80,81]. In recent years, COFs have been widely recognized by researchers as an excellent material that can be used for electrocatalytic CO_2_ reduction reactions due to their rich pore structure that can facilitate the adsorption and desorption of gas molecules during the reaction. By consciously introducing functional units such as bipyridyl, porphyrin, and phthalocyanine in the COF backbone to link metal ions for electrocatalytic CO_2_ reduction reactions, they have been widely explored in CO_2_ reduction reactions [82,83,84]. 

Recently, fluorinated COFs have also been effective catalysts for electrocatalysis. CTFs are a class of COF materials consisting of three elements, C, N, and H, and aromatic triazine units connected by covalent bonds. Since CTFs usually have a large specific surface area, high nitrogen content, excellent stability, and electrical conductivity, CTFs have a wide range of applications in gas separation and storage, energy storage, and electrocatalysis [85,86,87,88,89,90]. We summarize the catalytic performance of COF catalysts in CO_2_ electroreduction in Table 4. Dai et al. [91] synthesized a fluorinated covalent triazine skeleton (F-CTF-1) with the low-temperature ionothermal method using 2,3,5,6-tetrafluoroterephthalonitrile as a monomer (Figure 9a,b). The BET specific surface area of this fluorinated COF-containing material was 367 m^2^ g^−1^ and mainly microporous (Figure 9c–e). The introduction of fluorine enhanced the hydrophobic and pro-CO_2_ properties of F-CTF-1 and improved the interaction between F-CTF-1 and CO_2_ molecules. As an outcome, F-CTF-1 exhibited great Faraday efficiency (95.7%) at −0.8 V (vs. RHE) and was used for the electroreduction of CO_2_ to CO. Also with a microporous structure, Yaghi and colleagues [92] synthesized a 2D COF (COF-366-Co) in which the cobalt porphyrin site could catalyze the conversion of CO_2_ to CO by CO_2_RR. They synthesized a model framework (COF-366-Co) by the imine condensation of 5,10,15,20-tetrakis[(4-aminophenyl)porphinato]cobalt[Co(TAP)] with 1,4-benzenedicarboxaldehyde (BDA). The porous COF material was evacuated by activation with supercritical carbon dioxide and heating to 100 °C for 18 h. The surface area was determined to be 1360 m^2^ g^−1^. The adsorption branch fitted using density functional theory (DFT) showed a relatively narrow pore size distribution (10 to 18 Å), consistent with the proposed model. When tested in an aqueous solution, COF-336-Co showed a high Faraday efficiency of 90% Co (FE_CO_), which is 10% higher than that of the molecular cobalt porphyrin catalyst. It is therefore inferred that micropores would allow for a higher CO_2_ adsorption capacity in the framework, and the greater electrocatalytic activity and chemical availability of the catalytic cobalt porphyrin active site.

It was found that the difference in pore size in similar structures also affects the performance of COF catalysts. Recently, Huang et al. [93] constructed two catalysts, CuPcF_8_-CoNPc-COF and CuPcF_8_-CoPc-COF (Figure 10a), with similar pore structures. The adsorption curves of both COFs were classified as typical type I isotherms, which is characteristic of micropores (Figure 10c,d). The surface areas of CuPcF_8_-CoPc-COF and CuPcF_8_-CoNPc-COF were determined to be 376 and 452 m^2^ g^−1^, respectively. Correspondingly, their pore volumes were calculated to be 0.23 and 0.30 cm^3^ g^−1^. Non-local density flooding theory (NLDFT) was used to evaluate their pore sizes of 1.0 and 1.2 nm (Figure 6c,d). Meanwhile, the CO_2_ capacities of CuPcF_8_-CoPc-COF and CuPcF_8_-CoNPc-COF were 18 and 35 mg g^−1^, respectively. It can be seen that CuPcF_8_-CoNPc-COF has a larger pore size, specific surface area, and better CO_2_ capacity compared to CuPcF_8_-CoPc-COF, which is more favorable to improve the catalytic activity of CO_2_RR. Thus, when tested in 0.5 M CsHCO_3_, the FE_CO_ of CuPcF_8_-CoPc-COF reached a peak of 91% at −0.70 V (vs. RHE), while the FE_CO_ of CuPcF_8_-CoNPc-COF was as high as 97% at −0.62 V (vs. RHE).

Similarly, mesoporous structures are also present in COF materials, and Zeng et al. [94] formed COF on the surface of Mg/Al-LDH, and the obtained catalysts showed a layered and porous morphology with a thickness of about 36 nm and an abundance of active sites. The N_2_ adsorption isotherm at 77 K showed a BET surface area of 123 m^2^ g^−1^ for 2D-Co-COF500 (Figure 11a). The pore size distribution plot shows that the pore volume of 2D-CO-COF500 is 0.52 cm^3^ g^−1^, where most of the pores are between 25 and 80 nm in diameter (Figure 11b). Electrochemical tests were then performed on this material. The FE of CO was found to be 66.1% at −0.5 V, 84.0% at −0.6 V, 92.1% at −0.7 V, 96.5% at −0.8 V, 96.0 at −0.9 V, and finally 80.2% at −1.0 V (Figure 11c), while electrochemical tests on CO-COF500 revealed that the FEs of CO were all at the same test potential. Among them, 2D-Co-COF500 reached a maximum FE of 96.5% for CO at −0.8 V, and the catalytic performance was better than most of the current COF materials. To further investigate the catalyst activity, the CO current density (j_CO_) was tested. It was found that 2D-CO-COF500 had the highest current density of 17.9 mA cm^−2^ at a potential of −1.0 V (Figure 11d), while Co-COF500 also reached a maximum current density of 6.8 mA cm^−2^ at −0.9 V.

For materials with multi-level pore structures with both micro- and mesopores, the large specific surface area and rich pore structure allows them to have a strong CO_2_ adsorption capacity, enhanced mass transfer, and more exposed reactive sites, ultimately leading to stronger CO_2_ reduction performance. Zhang et al. [95] used porphyrin-based covalent organic framework (Por-COF) nanosheets vertically anchored to carbon nanotubes (CNT) by covalent linkage for an efficient electrocatalytic CO_2_ reduction reaction. The porosity structure of Por-COF and MWCNT-Por-COF-M were investigated by the BET test at 77 K. The results indicate that all the samples exhibit type I adsorption isotherms and the pore size distributions are concentrated on 15 Å (Figure 12a,b). The electrochemical test results show that the MWCNT-Por-COF-M (M: Co, Ni, Fe) material has higher electrocatalytic performance and Faraday efficiency than Por-COF-M. Specifically, MWCNT-Por-COF-Co achieves the highest Faraday efficiency of 99.3% for CO at -0.6 V, and the highest current density of 18.77 mA cm^−2^ at −1.0 V and TOF of 70.6 s^−1^ (Figure 12c). The electrochemical test results indicate that the excellent electrocatalytic activity of MWCNT-Por-COF-Co is due to the effective electron transfer. In addition, they used MWCNT-Por-COF-Cu in a flow cell for a CO_2_RR test with electrolytes in 1.0 M KOH. The test results illustrate that MWCNT-Por-COF-Cu has a higher CO_2_ catalytic efficiency than MWCNT-Por-COF, the product is mainly CH_4_, and the Faraday efficiency can reach up to 71.2% (Figure 12d). Sun et al. [96] used tetraanhydrides of 2,3,9,10,16,17,23,24-octacarboxyphthalocyanine cobalt(II) (CoTAPc) as a junction to couple with 5,15-di(4-aminophenyl)-10,20-diphenylporphyrin (DAPor) or 5,15,10,20-tetrayl(4-aminophenyl)porphyrin (TAPor) by imination reaction to fabricate novel coupled phthalocyanine–porphyrin type 1:2 (CoPc-2H_2_Por) or type 1:1 (CoPc-H_2_Por) COFs. Both type 1:2 and type 1:1 COFs were found to exhibit over 90% maximum Faraday efficiency and high stability in CO_2_ electrocatalytic reduction reactions. Meanwhile, CoPc-2H_2_Por has a larger pore size and a more conjugated structure than CoPc-H_2_Por, leading to more efficiency in electron transfer, more adsorption and reaction of CO_2_, and rapid proton transfer during the reaction, which ultimately leads to higher reaction kinetics, resulting in better CO_2_RR activity than CoPc-H2Por.

**Table 4 materials-16-01630-t004:** Summary of catalytic performance of COF catalysts in CO_2_ electroreduction.

Catalysts	E/V vs. RHE	Major Products	FE/%	Structure	Ref.
COF-366-Cu (HS)	−0.9	CH_4_	52.4	microporous	[86]
F-CTF-1-275	−0.8	CO	95.7	microporous	[87]
COF-366-Co	−0.67	CO	90	microporous	[92]
CuPcF8-CoNPc-COF	−0.62	CO	97	microporous	[93]
2D-Co-COF500	−0.8	CO	96.5	mesoporous	[94]
MWCNT-PorCOF-Cu	−1.0	CH_4_	71.2	micro- and mesoporous	[95]
CoPc-2H_2_Por	−0.55	CO	95	micro- and mesoporous	[96]
AAn-COF-Cu (NF)	−0.9	CH_4_	77	microporous	[97]
FN-CTF-400	−0.8	CH_4_	91.7	microporous	[98]
TPE-CoPor-COF	−0.7	CO	95	microporous	[99]
3D-Por(Co/H)-COF	−0.6	CO	92.4	microporous	[100]
COF@CoPor	−0.6	CO	94.3	microporous	[101]
NiPc-TFPN COF	−0.9	CO	99.8	microporous	[102]
CoPc-PI-COF-3	−0.9	CO	96	microporous	[103]
COF-366-(OMe)_2_-Co@CNT	−0.68	CO	93.6	macroporous	[104]
CTF-Cu	−1.47	C_2_H_4_	30.6	micro- and mesoporous	[105]

### 2.4. Nitrogen-Containing Polymers (N-Polymers)

There is also a class of porous polymeric materials called “metal-loaded ligand-active nitrogen-containing polymers” (N-polymers). This kind of N-polymer is characterized by the presence of nitrogen functional groups that complex metal cations or anchor metal nanoparticles within the polymer backbone (e.g., polyaniline and polypyrrole) or side chains (e.g., polyvinyl terbium pyridine and poly (4-vinyl pyridine)). The application of such materials in the field of CO_2_ electroreduction, however, dates back to three decades ago, when Wrighton’s [106] and Meyer’s groups [107] introduced N-polymers in the field of non-homogeneous CO_2_ electroreduction and found that such catalysts have excellent catalytic properties in CO_2_ conversion. So far, metal-doped N-polymer materials have been tested to synthesize polycarbon products up to propanol.

CO_2_ adsorption by porous polymers has been used as a competitive strategy for selective CO_2_ removal/capture [108,109,110]. Due to the rapid development of nitrogen-containing nanoporous polymeric materials, the application of N-polymers in CO_2_ capture has achieved remarkable success. Nitrogen-rich carbons are of great interest because of the contribution of N as a basic or polar site for effective interaction with CO_2_ [111,112]. Therefore, the preparation of nitrogen-rich carbon materials for CO_2_ capture is quite attractive and important in various applications such as CO_2_ capture. Due to the high porosity and relatively high nitrogen content, the adsorption of polyaniline (PANI) is very effective in both liquid and gas phases [113,114,115]. Park and his colleagues [116] prepared a porous carbon material using melamine and PANI with milder reaction conditions (≤650 °C). The prepared materials (MPCs) were found to have some CO_2_ adsorption capacity and therefore could be used for CO_2_ adsorption. It can be seen from Figure 13a that the adsorption capacity of PDCs for N_2_ increases with the increase in the pyrolysis temperature. Therefore, for N_2_ adsorption capacity, MPC-500 is the worst while MPC-650 is the best. The pore size distribution (PSD) of the material was also obtained from the isothermal curve shown in Figure 13a and shown in Figure 13b by nonlocal linear density functional theory (NLDFT). The adsorption capacities of the different materials prepared for CO_2_ and N_2_ are depicted from Figure 13c. As shown in Figure 13a, the adsorption of CO_2_ decreases according to the decrease in the pyrolysis temperature of the material. Thus, MPC-600 has the highest CO_2_ adsorption capacity and MPC-500 has the lowest CO_2_ adsorption capacity. Among them, MPC-550 showed superior CO_2_ selectivity, attributing this performance to the increase in the elemental N content of the material, the decrease in porosity, or the decrease in pore size and the increase in defects.

P4VP is an N-heteroaromatic polymer with a mesoporous structure that allows easy diffusion of small molecules and anions through the polymer network [117,118]. Nowadays, this N-polymer has been introduced into the electrochemical reduction of CO_2_ reactions, and Koper and colleagues [119] showed a significant improvement in the selectivity of polycrystalline Cu and Au electrodes for HCOOH at low overpotentials after chemical modification by using poly(4-vinyl pyridine) (P4VP) layers. Measurements showed that the hydrophobicity of the P4VP layer limited the mass transport of HCO_3_^−^ and H_2_O, but had little effect on the mass transport of CO_2_ due to its mesoporous structure.

Meanwhile, g-C_3_N_4_ is also a common N-polymer material with a structure in which the C and N atoms are sp2-hybridized to form a π-type conjugated system with a highly delocalized domain. Due to its high stability no matter the extreme chemical environment or high temperature, proper pore structure, redox potential, and abundant functional groups on the surface, there are also many studies based on g-C_3_N_4_ materials reported for energy and environmental applications [120,121], especially for electrocatalytic CO_2_ reduction [122]. Zhao et al. [123] prepared a Au-CDots-C_3_N_4_ material for CO_2_RR that can efficiently convert CO_2_ to CO. Electrochemical tests showed that 4 wt% Au-CDots-C_3_N_4_ has optimal CO_2_ electrocatalytic performance, and this material can undergo CO_2_ electrocatalytic reactions at −0.5 V and the Faraday efficiency of CO reached a maximum of 79.8%. The surface area of 4 wt% Au-CDots-C_3_N_4_ was investigated by the BET method at 77K. Figure 14a demonstrates the BET plot of 4 wt% Au-CDots-C_3_N_4_, which is consistent with a typical type IV isotherm with an H3-type hysteresis loop, indicating a large number of mesopores in this material. The specific surface area of this material was tested to be 117 m^2^ g^−1^, which originates from the rich void structure unique to the C_3_N_4_ matrix. Additionally, the pore size distribution obtained by the BJH method test can be shown by Figure 14a. Hu et al. [124] tried to modify Cu and Ru onto g-C_3_N_4_ (Cu_x_Ru_y_CN) and use it as an electrode for electrocatalytic CO_2_ reduction, and it can be seen from the test results that Cu and Ru modification on g-C_3_N_4_ can be an effective way to improve its electrocatalytic performance for energy applications. As can be seen in Figure 14a, the surface area of the g-C_3_N_4_ sample decreases with increasing Cu content, which indicates that the increased Ru and tiny CuO particles partially block the mesopores. Therefore, this result indicates that the Cu_x_Ru_y_CN samples with proper Cu and Ru ratios show the best BET surface area, pore volume and pore size under optimized Cu and Ru decoration to attract reactant molecules and provide active sites for better electrocatalytic reactions.

To summarize, the influence of the structural characteristics of four different types of porous polymer materials, POPs, PCPs, COFs, and N-polymers, on CO_2_ capture and electrocatalytic reduction have been reviewed. Their structural features and advantages and disadvantages for CO_2_ capture and electrocatalytic reduction are further summarized in Table 5.

## 3. Conclusions and Perspectives

Global climate change caused by excessive CO_2_ emissions has become a major concern in modern society. CO_2_ capture and conversion can remove excess CO_2_ from the atmosphere and reduce it into products with high added value. Porous polymers have been applied as potential CO_2_ adsorbents and electroreduction catalysts due to their high surface area, designable functionalization, tunable porosity, and various reaction sites. This review has systematically summarized the influence of the structural characteristics of different porous materials on CO_2_ capture and electrocatalytic reduction. Four different types of porous polymers, including POPs, PCPs, COFs, and N-polymers, have been discussed in depth. Porous polymers can serve as a multifunctional platform to achieve the desired purpose. Their high specific surface area and porosity enable them to absorb and store CO_2_. The design of the structure can not only improve the selective adsorption of CO_2_, but also allow the porous polymer as a catalyst to capture and convert CO_2_ into high-value-added chemicals. The adsorption of CO_2_ on porous polymers mainly depends on pore filling, so the highly developed porous features determine the ability of CO_2_ capture. It is worth noting that the introduction of heteroatoms can effectively improve the affinity of porous polymers to CO_2_ compared with other gases. Heteroatoms act as binding sites of CO_2_ through dipole–quadrupole interaction. In addition, the functionalization of the surface CO_2_-philic groups helps to further improve the adsorption capacity and selectivity. 

The role of pore structures in CO_2_ adsorption and electroreduction should be particularly emphasized. An abundant variety of building blocks with various functional groups are utilized for porous materials’ synthesis, which allows the pore structure and the chemical environment within the pore to be designed for a specific purpose. The microporous and ultramicroporous structures are considered to be preferred for the adsorption of CO_2_ molecules. The physical and chemical interactions between pore walls and CO_2_ molecules can be enhanced by fine-tuning the pore size and structure, which improves the catalytic efficiency of the electrocatalyst. The mesoporous structure is conducive to the increase in the accessibility of catalytic reaction sites, mass transfer, and the enrichment of reactants in the pores, which will accelerate the rate of the CO_2_ electrocatalytic reduction reaction. The hierarchical porous structure generally consists of a composite of micropores and mesopores. Although the surface area and pore volume of the polymers decrease as the mesopores increase, the disadvantage of single micropores that are not conducive to the diffusion of reactants and products at a single microporous pore size is overcome. Hierarchical porous polymers are promising candidates for expanding the electrocatalyst family. Effective electrocatalytic reactions occur at certain active sites. Therefore, it is of practical significance to create and regulate porous structures in order to expose more catalytic active sites. 

Despite the above progresses achieved, there are still some great challenges to overcome in the design and synthesis of porous polymers towards highly efficient CO_2_ capture and catalytic reduction. Promising future research directions in this field are also prospected as follows.

Firstly, the specific surface area, pore size, and pore morphology of porous polymer materials are tuned by functionalized modification of the polymer backbone to achieve the result of improved catalytic performance, for example, functional elements of a semiconductor nature can be introduced into the backbone structure, thus enabling porous polymer materials to achieve and improve performance in electrocatalysis, energy storage, etc. However, for now, achieving specific tuning of porous polymer materials is still a great challenge. More attention should be paid to boosting the coordination of active sites and pore topology to prevent the separation of metal ions or mixtures while improving the electrocatalytic performance, so as to provide long-term stability. Hierarchical porous structures can be constructed by developing advanced synthetic methods to improve the adsorption and activation of electrocatalysts for CO_2_. The chemical environment within the pores can be regulated by introducing heteroatoms, substituents, and metal components to stabilize key reaction intermediates and promote the charge transfer properties of porous polymers.

Secondly, the mechanism of porous polymers in the electrocatalytic reduction of CO_2_ is not well understood, especially for the exploration of the pathway of polycarbonate product generation. In the current study, different reaction intermediates are detected corresponding to different reaction mechanisms, but most of them stay in proposing the mechanism, and the investigation of experimental evidence is not comprehensive. It is difficult to accurately predict the electrocatalytic performance of porous polymers only by theoretical calculation, which limits the development of novel advanced porous polymers for CO_2_ electrocatalysts. Development of cutting-edge in situ monitoring technologies could bring about promising solutions to this issue. In situ surface characterization techniques, such as XPS, TEM, XRD, and XAS, can provide clear information about the evolution of the active sites of the electrocatalyst and the intermediate species of key reactions, which will deepen the understanding of CO_2_ capture and catalytic reduction by porous polymers. 

Third, the lab-scale research generally utilizes concentrated CO_2_ for testing, which is quite different from the atmospheric CO_2_ in practical situations. To facilitate the conversion of basic research achievements into products in industry, atmospheric samples are recommended to be used in basic research. In addition, most of the existing synthetic means are costly and have harsh reaction conditions, making it difficult to achieve low-cost mass production. We should focus on finding environmentally friendly, simple, and inexpensive co-production methods to promote the industrial application of porous polymer materials. 

In conclusion, porous polymers are highly efficient CO_2_ capture and conversion materials with great development potential. There are still severe challenges including lack of accurate design of porous polymers with well-coordinated catalytic active sites and pore topology, insufficient cutting-edge in situ monitoring technologies, as well as impractical testing conditions and high production costs. Through reasonable structure design, in-depth research on electrocatalysis mechanisms, and reduction of production cost, the practicability of porous polymers can be further promoted.

## Data Availability

Data sharing is not applicable to this article as no new data were created in this study.

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
