# Peer review of "Porous Polymer Materials for CO_2_ Capture and Electrocatalytic Reduction"

_materials, 2023, doi:10.3390/ma16041630_

Round 1

Reviewer 1 Report

The manuscript entitled “Porous polymer materials for CO2 capture and electrocatalytic reduction” presents a review CO2 capture in porous polymers. Reducing CO2 in the atmosphere is aimed globally. The topic is given review are promising and match with the scope of Materials. Following is my report and comments needed to be addressed.

1- The authors should conclude the section of optimal design of porous polymers for expected development in definite and direct points. 

2- The main products obtained after gas capture should be listed in a separate table.  

3- I recommend adding the following citations focusing on CO2 capture in environmental friendly composites:

https://www.sciencedirect.com/science/article/pii/S0950061820335856

https://link.springer.com/article/10.1007/s10924-022-02631-x

4- In Introduction, some phrases are very long than usual. 

5- What about the ability to capture CO2 during the curing/crosslinking of thermoset polymers, such as epoxy and polyester? Is that available?

Reviewer 2 Report

Manuscript ID: materials-2173065

Title: Porous polymer materials for CO2 capture and electrocatalytic reduction

Dear Authors

 This manuscript reports the development of advanced porous polymer electrocatalysts for efficient CO2 reduction. However, interesting results have been obtained, overall, the manuscript needs major revision before it could be accepted for publication in the Materials Journal. In this regard, the authors should improve their work according to the following indications.

1.        Both advantages and disadvantages of different porous polymer materials for CO2 capture and electrocatalytic reduction should be presented in a Table.

2.        It is appropriate that the authors add a section entitled important structural design features of materials for selective CO2 capture.

3.        A full CO2 capture process requires the design of an integrated capture and release process that enables efficient regeneration of the material and subsequent processing of CO2. In this regard, authors should provide some information about CO2 regeneration and their details.

4.        Could the author add a section entitled Post-synthetic modification and provide a comprehensive account of significant progress in the design and synthesis of porous polymer materials?

5.        The proposed mechanisms for CO2 capture and electrocatalytic reduction should be added and fully described.

Sincerely yours,

Round 2

Reviewer 2 Report

Manuscript ID: materials-2173065

Title: Porous polymer materials for CO2 capture and electrocatalytic reduction

The authors have not responded well to the mentioned comments by the referee, and therefore the current version is not acceptable for publication in this journal.

Sincerely yours,

Author Response

The authors thank the referee for the great effort in reviewing our manuscript. We have tried our best to respond to the five comments. Maybe because of our limited understanding on the referee's question, such response does not address well the referee’s concerns. But we think the current version is concise, and gives a clear and comprehensive review on POPs, PCPs, COFs, and N-polymers for CO2 capture and electrocatalytic reduction.